# Characteristics of Radio Frequency Dielectric Barrier Discharge Using Argon Doped with Nitrogen at Atmospheric Pressure

**DOI:** 10.3390/ma15217647

**Published:** 2022-10-31

**Authors:** Sen Li, Jiazhen Sun, Rui Sun, Jie Pan, Lin Wang, Chen Chen, Qiang Chen, Zhongwei Liu

**Affiliations:** 1Department of Information and Intelligent Engineering, Shanghai Publishing and Printing College, Shanghai 200093, China; 2State Key Laboratory of Biobased Material and Green Papermaking, Key Laboratory of Pulp and Paper Science & Technology of Ministry of Education/Shandong Province, Faculty of Light Industry, Qilu University of Technology (Shandong Academy of Sciences), Jinan 250353, China; 3Laboratory of Plasma Physics and Materials, Beijing Institute of Graphic Communication, Beijing 102600, China

**Keywords:** radio frequency, dielectric barrier discharge, atmospheric pressure

## Abstract

In order to study the characteristics of radio frequency dielectric barrier discharge (RF-DBD) using argon doped with nitrogen at atmospheric pressure, electrical and optical diagnoses of the discharge with different nitrogen ratios from 1% to 100% were carried out, and the self-organizing form of the filamentous plasma was studied through a transparent water electrode. At the same time, an ICCD camera was used to study the spatiotemporal evolution filamentous discharge during one cycle. Different from discharge using pure argon, using argon doped with nitrogen made the discharge change from glow discharge to filamentous discharge when the voltage increased to a certain value, and a higher nitrogen ratio made the filaments thicker and more sparsely arranged. Under different input power and nitrogen content conditions, several forms of glow discharge, hexagonal/irregularly arranged filamentous discharge and local filamentous discharge were obtained, all of which have potential applications to reduce the high cost of using inert gases.

## 1. Introduction

As is well known, plasma-processing technology has become a key technology in semiconductor, microelectronics, materials, medical and other industries and has shown its good economic and environmental benefits. Compared with the plasma produced by discharge under low pressure, the plasma produced by discharge at atmospheric pressure has the advantages of high production efficiency, realizing continuous production and low cost because it provides a chamber-free and low-cost route for industrial applications [1,2,3]. Among the various forms of atmospheric discharge [4,5,6,7,8,9,10,11,12], atmospheric pressure glow discharge (APGD) has become a research hotspot because of its volume and uniformity. In 1988, Okazak et al. reported a stable glow discharge achieved with inert gas under atmospheric pressure [13]. Then, Michael G. Kong’s team used inert gases to obtain a uniform and stable atmospheric glow discharge by adding a medium between flat RF electrodes; the discharge mechanism and mode of conversion of RF dielectric barrier discharge were studied deeply using the numerical simulation method [14,15,16,17,18]. Bin Li et al. obtained a stable glow discharge with a large gap of 5.5 mm using argon in RF dielectric barrier discharge [19]. The previous studies show that, compared with bare electrodes, dielectric barrier discharge (DBD) allows plasma to retain its large volume without constriction [15,16,20]. Among the various power sources used in atmospheric DBD discharge [9,21,22,23,24], RF power shows unique advantages in low breakdown voltage, stability and power density improvement [9,20,25,26]. On the other hand, filamentous discharge as a common discharge mode of DBD has also been studied a lot [27,28,29,30,31,32,33]. For example, Breazeal et al. observed hexagonal and stripe patterns in dielectric barrier discharges at near-atmospheric pressure [30]. Lifang Dong et al. studied the quadrilateral, hexagonal and concentric circles formed in filamentous discharge using middle-frequency power [31,32,33]. At present, gas used in atmospheric RF-DBD for studies on the mechanism and application are almost all expensive inert gases [20,34,35]. If non-inert gases such as nitrogen, or even air, as well as inert gas mixed with non-inert gas could be used for atmospheric RF-DBD application, the economic benefits of atmospheric RF-DBD will be further improved. Under the advantages of atmospheric pressure RF-DBD, in this paper, the characteristics of discharge using argon mixed with nitrogen (1–100%) are studied for potential applications using electrical and optical methods, as well as photos captured by an SLR camera using a transparent water electrode to study the discharge mode and self-organizing form of the discharge. An ICCD camera was used to study the spatiotemporal evolution of the filamentous discharge.

## 2. Experimental Facility

The experimental setup is shown in Figure 1: one copper electrode (10 × 10 cm^2^) marked as the RF electrode covered by a quartz sheet as a barrier layer (14 × 14 × 1 mm^3^) is connected to the power with a maximum power of 2 kW (13.56 MHz) through an automatic matching net (RF auto-match network PM-2K, Antorr, Beijing) to automatically convert the impedance (capacitive or inductive) of the plasma into a state that maximizes the power output of the RF source. The power supply can provide a breakdown voltage of about 2500 V at a 2.2 mm discharge gap. The upper electrode was designed as a water electrode, that is, a quartz box with a grounded circle sheet of copper inside and filled with tap water. Under such conditions, it works as an RF-DBD (double barrier layer). The bottom of the water electrode measured 8.9 mm × 10 mm^2^. All experiments were carried out under atmospheric pressure. The discharge gap between water electrode and the quartz sheet was fixed at 2.2 mm. Argon and nitrogen were used as discharge gas and the total flow rate of the gas mixture was fixed at 2 liters per minute (L/min). The current and voltage were measured with a wide-band current probe (Tektronix 6021AC, Beaverton, OR, USA) and a wide-band voltage probe (Tektronix P6015A, Beaverton, OR, USA). The probe signals were collected by an oscilloscope (Tektronix DPO 4104, Beaverton, OR, USA). Discharge photos were captured by an SLR camera (Nikon D7200) with a resolution of 6000 × 4000 pixels and an ICCD camera (Princeton Instrument-PI-MAX2) with a resolution of 1024 × 1024 pixels. The optical emission spectrum was obtained by a spectroscope (Avantes Avaspec 2048, Apeldoorn, The Netherlands) with a resolution of 0.12 nm.

In the experiment, different discharge gases entered the mixing chamber together, and then entered the discharge gap through a sieve plate to acquire a uniform and laminar gas flow. During discharge, two parameters were changed—the ratio of nitrogen in the mixture and the power supply through rotating the power supply knob—and the input power displayed on the automatic matcher was recorded, as well as the RMS voltage and RMS current of the discharge at each power.

## 3. Results and Discussion

### 3.1. Effect of Discharge Power on Discharge Characteristics

In the experiment, argon mixed with nitrogen was used as the discharge gas in the 2.2 mm double DBD. The flow rate of the gas mixture was sustained at 2 L/min with a different flow rate of nitrogen. After gas breakdown, the power was constantly increased by rotating the power supply knob at about the same angle, and the input power displayed on the automatic matcher was recorded as well as the RMS voltage and RMS current at each power. The I-V curve is shown in Figure 2. The four curves in Figure 2 correspond to discharge with different nitrogen contents of 20, 30, 40 and 50 mL/min. Figure 3 shows the discharge photos with a nitrogen content of 50 mL/min. It is obvious that the trend of each curve in Figure 2 is basically the same, which can be divided into three stages.

The first stage corresponds to points 1–3 in Figure 2 and Figure 3. The voltage first decreases, and then flattens out as the current increases, because in this process, the plasma has not filled the discharge gap, which can be seen from Figure 3. In the second stage, which corresponds to points 3–8 in Figure 2 and Figure 3, the voltage increases with the increase of the current, which shows a clear linear relationship and represents the uniform glow discharge with full plasma in the discharge gap. In the third stage, when the voltage increases to a certain value, both the current and the voltage decreased sharply, which is due to the change of discharge mode from glow discharge to filamentous discharge, as shown in Figure 3.

Therefore, it can be concluded that when the discharge voltage of argon doped with nitrogen reaches a certain value, the glow discharge changes into filamentous discharge, which cause the change in the plasma impedance and the current voltage value will be greatly decreased.

In order to study the pattern of the filamentous discharge, plasma photos were captured by an SLR camera vertically from the top of the water electrode, as shown in Figure 4. Figure 4a–d correspond to the last four points of the curve with a nitrogen content of 50 mL/min in Figure 2, and the power was 495 W, 550 W, 596 W and 656 W, respectively. It can be seen from Figure 4a (495 W) that the filaments did not fill the whole discharge area, and glow discharge exists in the lower left corner. Therefore, the discharge of Figure 4a can be regarded as a form of coexistence between glow discharge and filamentous discharge, in which filamentous discharge plays a dominant role. It can be easily observed that the filaments are arranged regularly as a hexagonal structure. As shown in Figure 4b, when the power was increased to 550 W, the area of glow discharge decreased significantly and a small number of filaments at the bottom split into a cluster of slimsy filaments, numbering about 3. Affected by this, the filaments of the nethermost part gradually lost the hexagonal structure. When the power was increased to 596 W (as shown in Figure 4c), the splitting of the filaments became more obvious, especially in the marginal region. The hexagonal structure formed by the filaments only exists in a few small areas. When the discharge power was further increased to 656 W, almost all of the filaments existed as clusters, most of which were composed of 3–5 slimsy filaments and the regular arrangement of hexagons completely disappeared. In this process, the voltage changed little as the current increased. It is concluded that the current transmitted by the filaments is much higher than that of the glow discharge, and the voltage is the direct cause of the formation of filaments rather than power. This means the voltage would increase as the power increases if the number of filaments does not change; however, the increased voltage brings the emergence of new filaments, which makes the voltage lower in reverse. For the large number of filaments, the curve shows a nearly horizontal trend.

Therefore, it can be concluded that double RF-DBD using argon doped with nitrogen could form a hexagonal self-organizing structure. A higher power makes the individual filaments change into clusters and lose their hexagonal structure.

### 3.2. Effect of Gas Ratio on Self-Organizing Structure of Filamentous Discharge

It has been concluded that the higher the nitrogen content, the more difficult the breakdown of gas. In order to study the influence of nitrogen content on the self-organizing structure of filamentous discharge, vertical photos were captured while the nitrogen flow rate was gradually increased under the same power of 420 W. The flow rate of the nitrogen was 20, 80, 150, 300, 600 and 1200 mL/min, respectively, while the flow rate of the gas mixture was constant at 2 L/min. It can be seen that filamentous discharge and glow discharge coexisted when the amount of nitrogen was small. When the flow rate of the nitrogen was increased to 150 mL/min, the glow discharge basically disappeared. When the flow rate of the nitrogen was 20 mL/min, the filaments kept moving irregularly and could not form a regular structure. When the flow rate of the nitrogen increased above 80 mL/min, obvious hexagonal structures were observed. When the flow rate of the nitrogen was above 300 mL/min, the number of filaments decreased rapidly, but the arrangement of the remaining filaments was still relatively aggregated and formed a hexagonal structure. In this process, the color of the filaments also changed accordingly. As the nitrogen ratio increased, the color gradually changed from light magenta to deep purple.

In addition, the diameter of the filaments and the distance between the centers of the two adjacent filaments were estimated by the photos from Figure 5. The diameters of the discharge wires are 0.1, 0.22, 0.27, 0.35, 0.41 and 0.52 cm, respectively, and the centers of the two adjacent filaments are 0.42, 0.53, 0.67, 0.72, 0.83 and 1.12 cm, respectively. The diameter and distance of the filaments increased gradually as the nitrogen content increased.

At 420 W, the plasma is extinguished when the nitrogen content is more than 60%. In order to obtain discharge at a higher nitrogen ratio, the power was further increased. Back to Figure 4d, at the power of 656 W, the filaments were already in a split state without a regular self-organizing pattern. On this basis, the nitrogen content was increased to more than 60% until 100%, and the discharge was maintained stably, as shown in Figure 6. By comparing the results in Figure 4 and Figure 5, it can be concluded that the nitrogen content must be between about 2.5% and 60% to obtain a regular hexagonal structure in filamentous discharge mode. In addition, a high nitrogen content cannot fill the entire discharge area, and a large area of filamentous discharge of hexagonal structure needs to meet the nitrogen content between about 2.5% and 10%. It is worth mentioning that, in this case, as the discharge filaments were moving slowly with the airflow body, plasma could be used in material surface treatment in a relatively uniform way. In addition, the higher the nitrogen content, the slower the discharge filaments move. When the nitrogen content exceeds 30%, the discharge filaments basically stop moving. In this case, the stationary regularly arranged discharge pattern has potential applications, such as plasma photonic crystals or the selective treatment of material surfaces.

### 3.3. Diagnosis of Emission Spectra

Figure 7 shows the emission spectrum of plasma with a nitrogen flow rate of 50 mL/min at 293 W. Table 1 shows the parameters of the three spectral lines marked in Figure 7. The change of intensity of the three spectral lines with power is given in Figure 8, in which the 12 points of each curve correspond to the curve of the nitrogen flow rate of 50 mL/min in Figure 2. It should be emphasized that when the discharge transformed into filamentous discharge, the light collected by the probe of the emission spectrometer only represents the average luminescence intensity of the whole plasma, rather than just filaments. It can be seen from Figure 8 that the intensity of each spectral line increased with power before the transition of the discharge mode. The difference is that the intensity of N357 and Ar772 rises faster at a higher power. This indicates that the number of corresponding species is produced faster at higher powers. After the change of the discharge mode, the intensity of the Ar772 spectral line continued to increase with power. However, the intensity of the two lines of nitrogen decreased sharply, and then showed a steady trend.

This indicates that the transition from glow discharge to filamentous discharge has no obvious effect on argon ionization. Combined with the photos in Figure 3 and Figure 4, argon ionization always exists in the form of bulk ionization caused by an electron avalanche [20]. The two nitrogen curves vary with power in much the same way as the voltage trend before and after the mode transition. On the one hand, the transition resulted in a significant change in the ionization of nitrogen, while on the other hand, the ionization was concentrated on the dielectric surface within the discharge filaments.

### 3.4. Diagnosis of Filamentous Discharge by ICCD Camera

In order to study the mechanism of filamentous discharge in atmospheric double dielectric discharge using argon-doped nitrogen, an ICCD was used to diagnose the plasma in the experiment. Figure 9 shows the ICCD photo of filamentous discharge with a gas mixture flow rate of 2 L/min and a nitrogen flow rate of 150 mL/min at 420 W. The exposure time was 100 μs and the parameters of the plasma whose vertical photo is shown are provided in Figure 5c. The plasma is obvious under the filamentous mode, and the luminescence is mainly concentrated on two sides of the up and down.

The ICCD camera was triggered by the signal of the current when it crosses zero forward, and four photos within one discharge cycle (73.7 ns) were obtained, as shown in Figure 10. The single exposure time was 2 ns, and the four photos corresponded to 0 T, 1/4 T, 1/2 T and 3/4 T of the discharge cycle. In the first half-cycle, during which the upper water electrode is a cathode, positive ions are forced upward to reach the cathode, resulting in a large amount of wall charge accumulated on the surface of the water electrode at 1/4 T. Since these wall charges are all positively charged, repulsive forces are generated between each other, and eventually the filaments form a balanced structure as a hexagonal shape. The second half of the period is consistent with the first. With the increase of power, more and more charges accumulated on the surface of the cathode, which made the inner repulsive force of the wall charges generated by one single filament become larger and larger, and finally split into a cluster of multiple roots.

In addition, it can be observed from the photos in Figure 4 and Figure 5 that, except for the filaments in Figure 5a, which are in a state of random migration, the whole plasma slowly moves down with the flow of the discharge gas, and the positions of the filaments are relatively fixed. On the other hand, it can be seen from the waveforms shown in Figure 11 that the current waveform of RF-DBD is smooth, which is different from the current waveform of middle-frequency DBD [23]. Therefore, it can be concluded that the filaments are not extinguished during one period.

Therefore, it can be inferred that the transformation of glow discharge to filamentous discharge in double dielectric discharge using argon-doped nitrogen is due to the fact that when the voltage reaches a certain value, ions gain enough speed to destroy the plasma sheath, and charge accumulation is formed on the cathode surface, thus forming filamentous discharge.

## 4. Conclusions

Based on the experimental results and analysis, RF-DBD using argon doped with nitrogen at atmospheric pressure can be sustained with a nitrogen content from 1% to 100%. The discharge modes include glow discharge, filamentous discharge and the coexistence of both. In filamentous discharge, the ionization and charge accumulation on the surface of the cathode medium are the main reasons for the formation of the filaments. Different from pure argon, using argon doped with nitrogen made the discharge change from glow discharge to filamentous discharge when the voltage increased to a certain value. The further increase of power made the original single filament split into a cluster of discharge channels composed of 3–5 filaments. When the nitrogen content was low, the glow discharge still existed after the change of discharge mode to filamentous discharge. As the nitrogen content increased, the glow discharge composition became lower until only the discharge filaments remained. A higher nitrogen ratio made the filaments thicker and more sparsely arranged. The filaments of filamentous discharge can form regular hexagonal structures under certain conditions, but it is difficult for the regular filamentous discharge to fill the whole discharge gap when the nitrogen exceeds 10%. Therefore, the potential applications of RF-DBD using argon-doped nitrogen could be divided into the following situations. First, glow discharge may be applied in uniform material surface treatment when the nitrogen content is low. Under filamentous discharge mode, no matter whether the filaments are arranged regularly or not, as long as the filaments occupy the discharge gap and move as a whole with the air flow, this can also be considered as a relatively uniform surface treatment method over a large area. When the nitrogen content exceeds 30%, the stationary regularly arranged discharge pattern has potential applications such as plasma photonic crystals and the selective treatment of material surfaces. Finally, when the nitrogen content is more than 60%, use is still possible in applications where discharge uniformity is not required.

## Figures and Tables

**Figure 1 materials-15-07647-f001:**
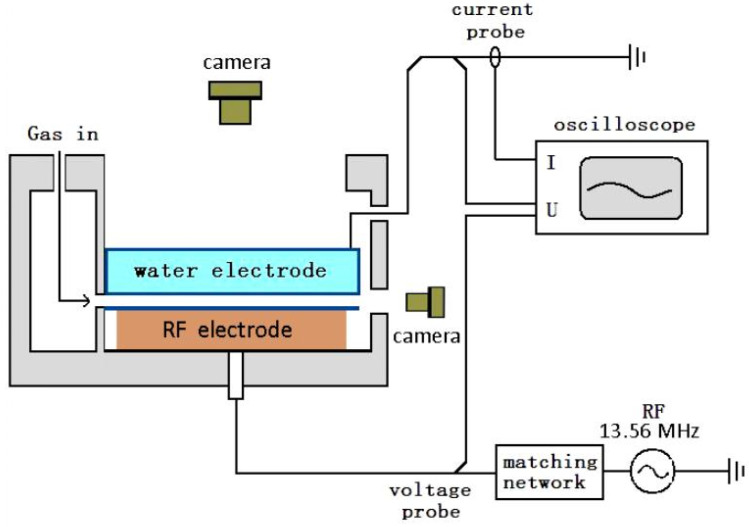
Schematic diagram of experimental setup.

**Figure 2 materials-15-07647-f002:**
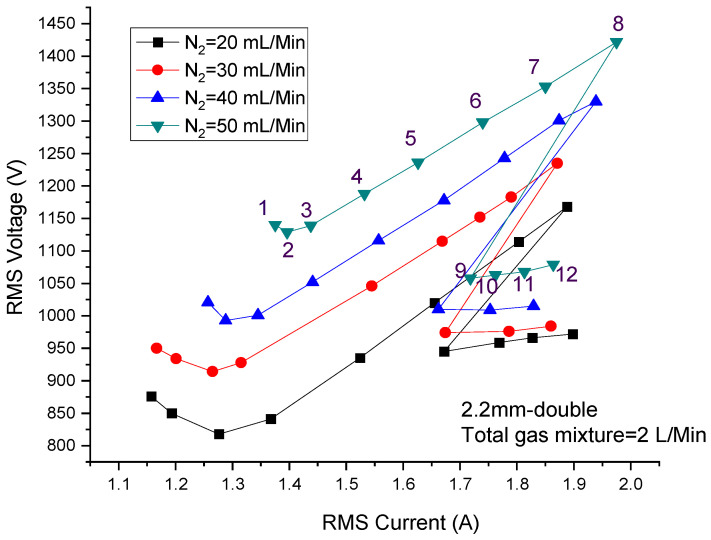
Current voltage curve of discharge using argon-doped nitrogen.

**Figure 3 materials-15-07647-f003:**
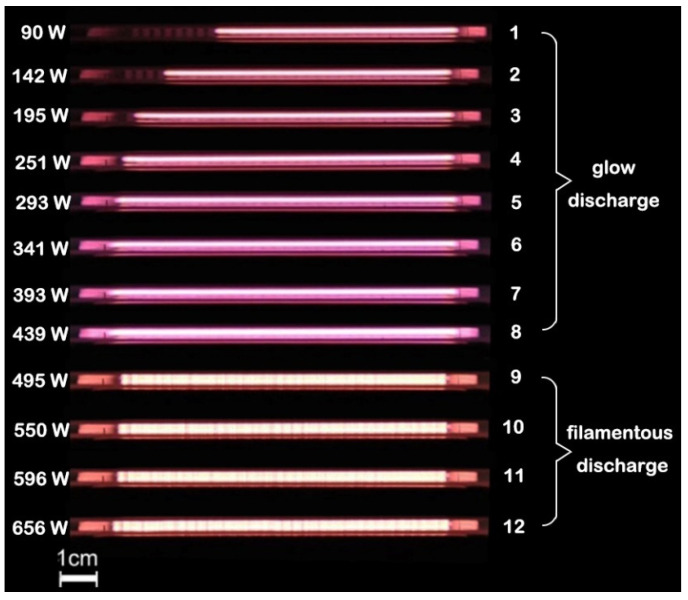
Photos of 2.2 mm RF-DBD using argon-doped nitrogen, N_2_ = 50 mL/min, gas mixture = 2 L/min. Photos 1–12 correspond to points 1–12 in Figure 2. Exposure time: 1 ms.

**Figure 4 materials-15-07647-f004:**
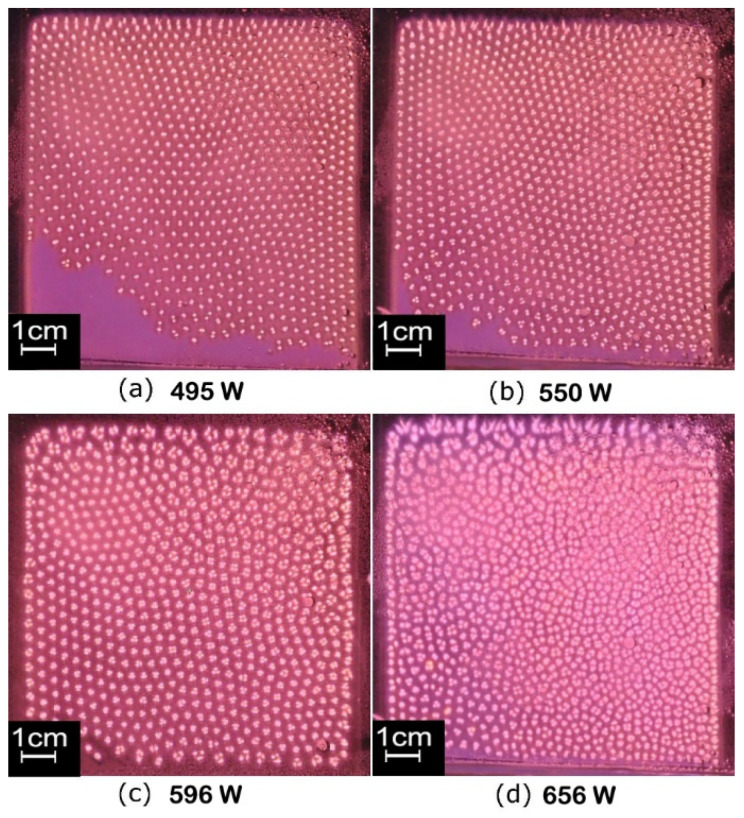
Effect of discharge power on self-organizing form of plasma, N_2_ = 50 mL/min, gas mixture = 2 L/min (N_2_ = 2.5 %). (**a**–**d**) correspond to points 9–12 in Figure 2 and photos 9–12 in Figure 3. Exposure time: 1 ms.

**Figure 5 materials-15-07647-f005:**
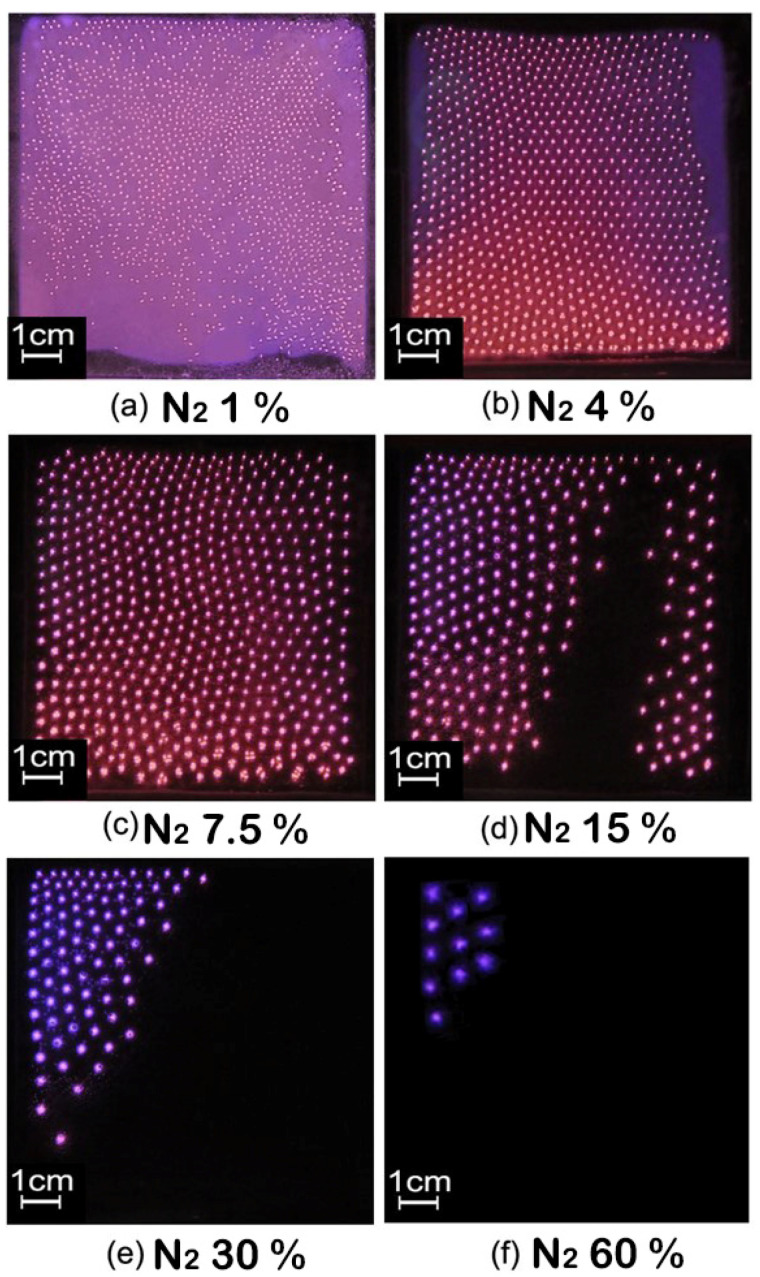
Effect of nitrogen content on self-organizing form of plasma. Exposure time: 1 ms.

**Figure 6 materials-15-07647-f006:**
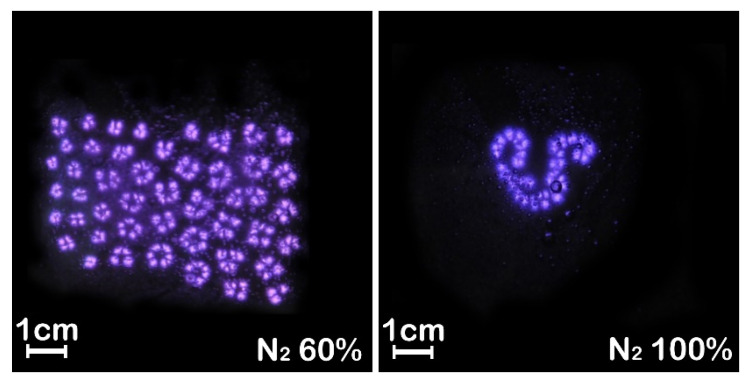
Photos of discharge with 60% and 100% nitrogen at 656 W. Exposure time: 1 ms.

**Figure 7 materials-15-07647-f007:**
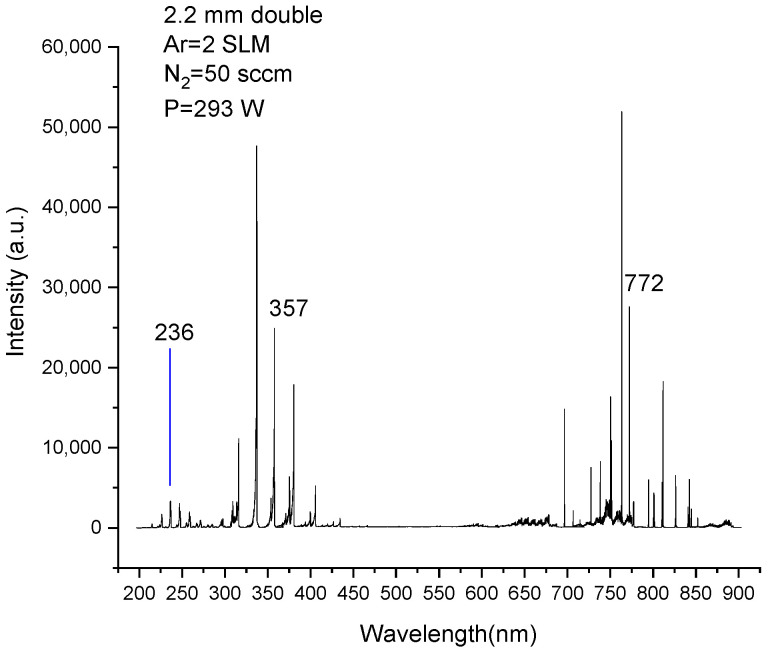
Emission spectra of discharge using argon-doped nitrogen, 236 nm: N_2_ (*A*^3^∑_u_^+^–X^1^∑_g_^+^), 357 nm: N_2_ (*C*^3^Π_u_, 0–*B*^3^Π_g_, 1), 772 nm: Argon (2p_7_–1s_5_).

**Figure 8 materials-15-07647-f008:**
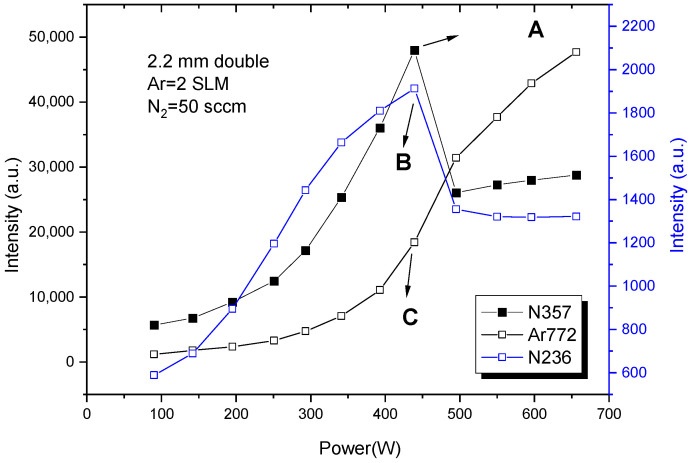
Variation of line intensity with power in discharge using argon-doped nitrogen. Points A, B and C in the three curves are the locations where the discharge mode changed from glow discharge to filamentous discharge.

**Figure 9 materials-15-07647-f009:**
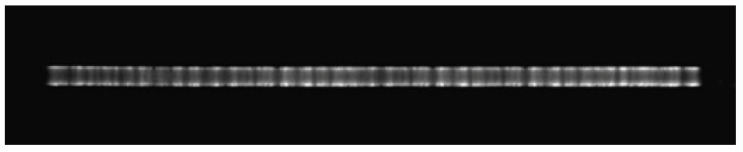
ICCD photo of filamentous discharge, P = 420 W, N_2_ = 150 mL/min, gas mixture = 2 L/min. Exposure time: 100 μs.

**Figure 10 materials-15-07647-f010:**
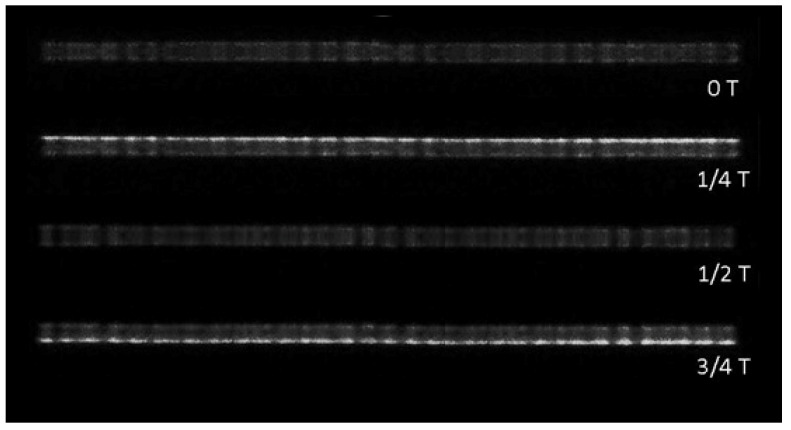
Photos of filamentous discharge at 0T, 1/4T, 1/2T and 3/4T of one current cycle (73.7 ns), P = 420 W, N_2_ = 150 mL/min, gas mixture = 2 L/min. Exposure time: 2 ns.

**Figure 11 materials-15-07647-f011:**
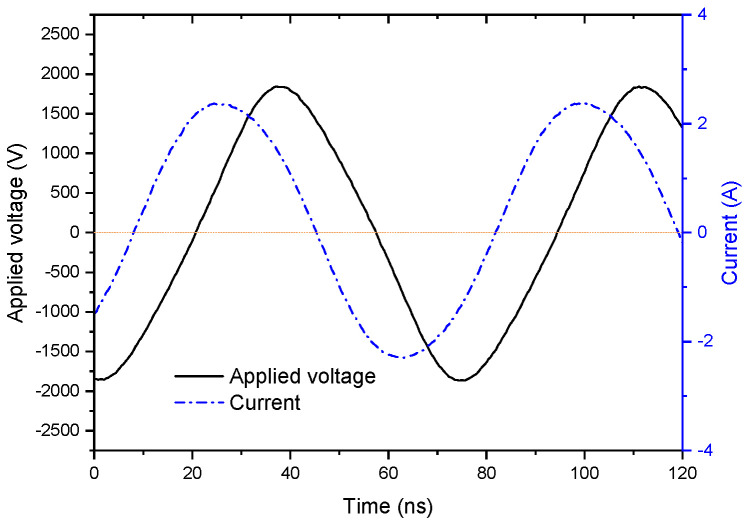
Waveforms of current and voltage, P = 420 W, N_2_ = 150 mL/min, gas mixture = 2 L/min.

**Table 1 materials-15-07647-t001:** Parameters of the argon and nitrogen spectral lines studied in the experiment.

Transition	*λ*/nm	Configurations	*A*_ki_/10^7^s^−1^	*E_i_*–*E_k_* (cm^−1^)
N_2_(*A*^3^∑_u_^+^, 0–X^1^∑_g_^+^, 3)	236	2*s*^2^*p*(3P°)3*d*-2*s*2*p*(^3^P°)4*p*	9.12	336,206.9–378,432.8
N_2_ (*C*^3^Π_u_, 0–*B*^3^Π_g_, 1)	357	2*s*^2^2*p*(^2^P°)3*p*-2*s*^2^2*p*(^2^P°)4*s*	1.21	168,892.21–196,711.54
Ar (2*p*_7_–1*s*_5_)	772	3*s*^2^3*p*^5^(^2^P°_3/2_)4*s*-3*s*^2^3*p*^5^(^2^P°_3/2_)4*p*	0.518	93,143.7600–106,087.2598

## Data Availability

Not applicable.

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
