# Peer review of "Characteristics of Radio Frequency Dielectric Barrier Discharge Using Argon Doped with Nitrogen at Atmospheric Pressure"

_materials, 2022, doi:10.3390/ma15217647_

Round 1

Reviewer 1 Report

The paper is devoted for characteristic of radio frequency dielectric barrier discharge investigations. Topic is generally interesting, however the paper contain unexplained places (below) and need major revisions.

The aim of the paper should be more clearly formulated.

Figure 7 should be more commented.

Numbers and measurements units should be written separately, for example line 86 should be ‘’495 W’’, not ‘’495W’’.

Conclusions should be rewritten in more informative way.

The reference list should be expanded.

Reviewer 2 Report

It lacks an analysis, it is necessary to go beyond: it goes up, it goes down.
What do you want to show in this paper?
No European or American quotation, although there has been a lot of work on this theme, cf. JP Boeuf

Reviewer 3 Report

Good: The concept of economic benefits of using nitrogen gas is of scientific and industrial importance.

To be improved: The observation results and discussion limit to be phenomenological so that the novelty of this research is very ambiguous. The list which can be improved is as follows.

1. on comparison with past studies to provide more scientific discussion

The following two results reported here seem NOT new in the current manuscript. You ignore the achievements of past DBD studies when discussing them. You can explain these results more deeply with comparing to past studies to highlight your novelty.

1-1. High discharge voltage in the presence of nitrogen (In general, molecular gases like nitrogen are already well known to require more voltages to maintain the plasma than rare gases. )

1-2. Hexagonal structures (This phenomenon has been deeply studied from the physical point of view by many researchers.)

2. on relationships between figures

You should specify which point in Fig. 2 corresponds to each photograph shown in Figs. 3 and 4. Because only the wattage is written for the photographs, we cannot get the relationships.

3. on current waveform

Could you show the current waveforms of the spiky DBD current? You can explain the result of Fig. 9 more clearly by referring such current waveforms.

Thank you for your effort.

Reviewer 4 Report

Referee’s report on the paper entitled:

“Characteristics of radio frequency dielectric barrier discharge using argon doped with nitrogen at atmospheric pressure” by Li et al.

Manuscript reference: 1965749

The submitted manuscript presents an experimental study about the characteristics of radio frequency dielectric barrier discharge produced on an atmosphere of argon doped with nitrogen. The authors performed electrical and optical diagnosis of the discharge for different gas ratios. I think the manuscript does not have minimum standards to be published, it lacks information and demonstrates a lack of interest from the authors on the preparation of the manuscript. The justification behind my opinion is exposed in the following points:

1)       The introduction of the manuscript is very short and weak, does not contain enough information to justify the importance of the topic, does not explain succinctly the background behind the study, neither presents a state-of-the-art about the works previously performed on the topic. This reveals the authors do not have minimum knowledge about the state-of-the-art and the works published on the topic. By this way, the authors completely fail on the objective of engaging the readers with the work they expose. In the second sentence of the introduction the authors cite 8 references, does not make any sense that they need to cite 8 works to support a single statement. In the third sentence the authors refer that “lot of studies” have been performed, however it seems none of them is worthless to be excruciated along the introduction. Authors do not expose results, or conclusions achieved by other authors and, thus, completely fail on connecting their work and what exists in the literature.

2)       Experimental setup section is also quite short and lacks the procedure explanation and several equipment characteristics. The authors do not mention the model of the power source, just the power and frequency… what are the levels of applied voltage achieve by this power source. This power source is connecting to a matching network, what is the purpose of the matching network? It is constituted by what? Usually, matching networks are composed of capacitors and inductors, what are the specifications of these components?  They use two cameras what is the resolution of these cameras? What is the exposure time of the cameras? When are the shots taken? All of this information is missing. A systematic description of the procedure used in order to obtain the results is also missing.

3)       Figure 2, the units on the concentration of N2 showed in the legend are missing.

4)       Figure 3 shows several photos what is the difference between them? It is not identified in the figure… Was each photo taken for a different applied current?

5)       In page 3, second paragraph authors write: “In the first stage, the voltage decreases with the increase of the current…” what do you mean by first stage? Currents from 1.15 amps to 1.3 amps? But what to you set in the experiment? Do you stablish the current or the voltage? I would say that usually the voltage is applied to the DBD and the current is a consequence of the applied voltage, but since the experimental section lacks a description of the procedure it is very hard to follow the presented results.

6)       In the end of the paragraph 2 of page 3, the authors write: “… which is due to the change of discharge mode from glow discharge to filamentous discharge which could be easily observed from Fig. 3.” It cannot be easily observed from figure 3 since the readers don’t know which photo they should look at… Shall they guess?

7)       Figure 4 presents the effect of discharge power on self-organizing form of plasma, however the different powers of each subfigure are not identified in the figure. The same in Figure 5, that does not identify what is the discharge ratio for each subfigure.

Concluding, the manuscript lacks several information, and because of that becomes very hard to follow the results and discussion. The topic of research is interesting, but in my opinion the manuscript should be rejected. The authors should rewrite the manuscript in a proper way and resubmit.  

Round 2

Reviewer 1 Report

Authors make proper corrections according to reviewer remarks and I suggest to publish the paper as it is.

Reviewer 2 Report

Thanks for the changes that improve the document, however the analysis remains weak.

Reviewer 3 Report

Thank you for your revision.

Reviewer 4 Report

Authors did a great effort on improving the manuscript and considered all my concerns. I believe the quality of the manuscript was significantly improved and can be now accepected for publication.